# DFEN: Dual Feature Enhancement Network for Remote Sensing Image Caption

**Weihua Zhao** [1], **Wenzhong Yang** [1,2,*], **Danny Chen** [1] **and Fuyuan Wei** [1]

1   School of Information Science and Engineering, Xinjiang University, Urumqi 830017, China
2   Xinjiang Key Laboratory of Multilingual Information Technology, Xinjiang University, Urumqi 830017, China
*   Correspondence: yangwenzhong@xju.edu.cn

**Abstract:** The remote sensing image caption can acquire ground objects and the semantic relationships between different ground objects. Existing remote sensing image caption algorithms do not acquire enough ground object information from remote-sensing images, resulting in inaccurate captions. As a result, this paper proposes a codec-based Dual Feature Enhancement Network ("DFEN") to enhance ground object information from both image and text levels. We build the Image-Enhancement module at the image level using the multiscale characteristics of remote sensing images. Furthermore, more discriminative image context features are obtained through the Image-Enhancement module. The hierarchical attention mechanism aggregates multi-level features and supplements the ground object information ignored due to large-scale differences. At the text level, we use the image's potential visual features to guide the Text-Enhance module, resulting in text guidance features that correctly focus on the information of the ground objects. Experiment results show that the DFEN model can enhance ground object information from images and text. Specifically, the BLEU-1 index increased by 8.6% in UCM-caption, 2.3% in Sydney-caption, and 5.1% in RSICD. The DFEN model has promoted the exploration of advanced semantics of remote sensing images and facilitated the development of remote sensing image caption.

**Keywords:** remote sensing image caption; image context features; text guidance features

## 1. Introduction

Image captioning [1] enables the computer to automatically generate a caption based on a specified image, allowing image data to be converted to text data and providing a more advanced understanding of the image. Blind navigation, image translation, voice assistants, and image retrieval all use this technique. Because of advances in remote sensing technology, computer vision tasks such as scene classification and object detection [2] can now obtain scene categories and ground objects from remote sensing images, but not semantic relationships among ground objects. Remote sensing image caption [3] can recognize semantic relationships between ground objects and generate accurate and natural captions. The remote sensing image caption is critical in many scenarios, including remote sensing image retrieval, disaster reports, and urban change reports.

CNN (Convolutional Neural Network) can extract rich and discriminative image features in computer vision. Qu et al. [4] were the first to use CNN combined with sequence models in the remote sensing image caption. Lu et al. [5] first used the attention mechanism to align the image and caption. To obtain more robust remote sensing image captions, Zhang et al. [6] proposed using the fully connected layer as image attribute information to bridge the "semantic gap" between low and high-level semantics. Huang et al. [7] proposed image feature denoising to aggregate multiscale image features. These methods use CNN to obtain visual features of the image. However, the CNN will lose some ground object information after multiple pooling. Moreover, the large-scale change of remote sensing images makes it difficult for CNN to recognize ground objects correctly. These are the reasons

for insufficient information acquisition of ground objects. Furthermore, the remote sensing image caption requires not only visual information but also text information. Inaccurate ground object information results in incorrect semantic expression and misinterpretation of remote sensing images.

To address the abovementioned limitations, the DFEN model proposes using the Image-Enhance module's hierarchical attention to obtain more discriminative image context features. Moreover, based on the ground objects in the image, the Text-Enhance module's gating mechanism is used to obtain the text guidance features of the whole interaction between images and texts and guide the DFEN to pay attention to the text information of the ground objects correctly. The problem of inaccurate captioning of remote sensing images caused by the insufficient acquisition of ground object information is solved using the dual features of ground object information.

In summary, the following are the paper's contributions:

- This paper proposes a DFEN model accurately representing ground object information in remote sensing images. DFEN is the first model to improve ground objects from image and text levels;
- The Image-Enhance module is used in this paper to obtain more discriminative image context features in order to enrich the visual representation of image ground objects from an image level. The Image-Enhance module facilitates the representation of visual features of the remote sensing images;
- The Text-Enhance module is used in this paper to obtain text guidance features with high graphical interaction to guide the model to focus on the correct ground object information and achieve accurate ground object information focus from a text perspective. The Text-Enhance module combines remote sensing image caption and multi-label classification tasks.

## 2. Related Work

Remote sensing image caption generates captions based on the semantic information in remote sensing images. An important aspect of remote sensing image captioning is obtaining sufficient ground object information. Many researchers at this stage provide information for remote sensing images by performing other visual tasks. Anderson et al. [8] used an object detection technique to obtain an "attention region", which allows the attention mechanism to focus on the image's corresponding object. Wang et al. [9] used the Faster RCNN model to obtain the object, patch, and global features of remote sensing images to achieve the global representation of image ground objects to obtain the semantic information of tiny ground objects. Although the combination of object detection [10–13] and remote sensing image captioning can acquire visual features of the image adequately, visual features of the image acquired using object detection techniques contain much repetitive information. Zhao et al. [14] built structured attention mechanisms using semantic segmentation [15] techniques. Structured information can build local information effectively, but the selective search in semantic segmentation methods limits the number of region proposals. Lu et al. [16] thought sound can represent the attention of different observers and describe images more precisely. Although sound is a direction for multimodal development, there is little correlation between sound and image content. The RTRMN [17] model extracts semantic topics from the captions corresponding to remote sensing images as additional information about the images. Although topics can help the model generate detailed captions, the topics searched during testing were too fixed, and the visual consistency was poor. The SD-RSIC [18] model used summaries to provide more detailed information. The summary is another kind of complex textual information, and the time and space consumption by introducing complex text is also a waste.

Extensive experiments on remote sensing image captions reveal that the captions mainly comprise nouns, the names of ground objects in remote sensing images. The image multi-label classification task labels are similar to this noun information. As a result, this paper proposes using a pre-trained image multi-label classification model to obtain

entity features of image ground objects and complementing them with more discriminative image context features to enhance ground object information jointly. This also encourages interaction between remote sensing image caption and another fundamental task.

## 3. Proposed Method

This paper proposes the Dual Feature Enhancement Network (DFEN) that employs Image-Enhance and Text-Enhance modules to achieve the feature of ground object enhancement in both the image and text perspectives.

The remote sensing image caption task generates a caption composed of ordered word sequence $Y = [y_0, y_1, y_2, ... y_L]$ according to the given image $I$, $L$ is the sentence length, $\theta$ is the model parameter, which maximizes the probability $p(Y|I;\theta)$ of the caption:

$$\theta^* = argmax_\theta \sum_{(I,Y)} \log^{p(Y|I;\theta)} \tag{1}$$

The overall structure of the model is shown in Figure 1. The encoding part consists mainly of the ResNet model to obtain global features of the image. The obtained image features are fed into the Image-Enhance module to obtain sufficient visual features. The decoding part consists mainly of a Text-Enhance module and a two-layer LSTM module, which generates captions that match the semantic information of the image. $X_t$ is the input to the Attention-LSTM, which mainly consists of the average $\bar{v}$ of the global semantic features $v^1$ of the image, the word to be input at the current moment $w_t$ and the hidden states $h_{t-1}^1, h_{t-1}^2$ of the two LSTMs at the previous moment, i.e., $X_t = [\bar{v}, w_t, h_{t-1}^1, h_{t-1}^2]$. The global semantic information $v^1$ of the image is obtained from the different visual features $f_i$ of the image through pooling and concatenation operations, respectively. The Text-Enhance and Language-LSTM modules process the Attention-LSTM output to generate words up to the end-of-word flag <BOS>.

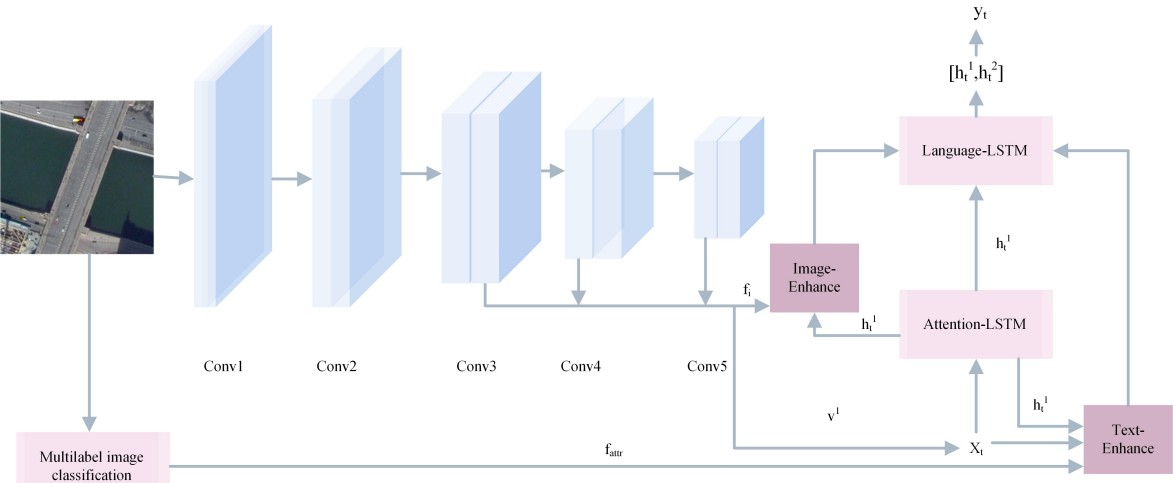

**Figure 1.** Overview of the proposed method—DFEN. The duality of DFEN is most visible in the visual feature of image acquisition and text information enhancement. The Image-Enhance module, in particular, captures global image context features using multiscale image features and the hidden state. The Text-Enhance module captures the image potential ground objects using the multi-label image classification module to achieve text information enhancement.

### 3.1. Image-Enhance Module

This paper proposes the Image-Enhance module for displaying more ground object information of remote sensing images. The Image-Enhance module extracts the image visual features of conv3, conv4 and conv5 in the ResNet model, respectively, and models the attention mechanism for the different visual features $f_i$ using a hierarchical attention

mechanism. Each image's visual feature contains unique information due to the different convolutional kernels. Using the hierarchical attention mechanism, the Image-Enhance can fuse multiple features to obtain the image context feature $v$. Figure 2 depicts the module's structure. The specific procedure is as follows:

$$v_i = f_i^a \times \delta(W_q Relu(W_{qv} f_i^a + W_{qh} h_t^1)) \tag{2}$$

$$v = [v_i] \tag{3}$$

$f_i^a$ is the $i$th visual feature filtered by the self-attention mechanism, and $h_t^1$ is the output hidden state of the Attention-LSTM; $\delta$, *Relu* denotes the activation function of the model; $W_q, W_q v, W_q h$ denotes the learned parameters; $[;]$ denotes the concatenation operation. Through the hierarchical attention mechanism, Equation (1) generates image features $v_i$ containing different ground object information, and Equation (2) concatenates different image features $v_i$. Both large and small ground object information in remote sensing images are included in $v$ via the Image-Enhance module.

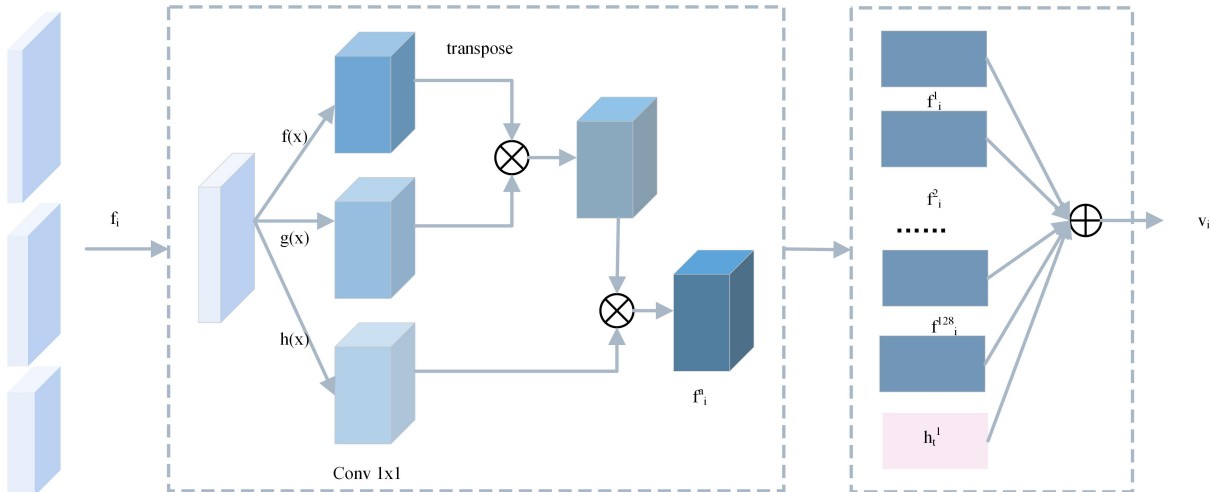

**Figure 2.** Overview of the Image-Enhance module. This module's hierarchical attention mechanism is primarily made up of two distinct attention mechanisms. The self-attention mechanism is the first layer of the attention mechanism, and it uses pixels at all positions in different visual features $f_i$ to obtain detailed information $f_i^a$, as shown in the first dotted box. The second layer of the attention mechanism is additive attention to aggregate image information related to text, as shown in the second dotted box. The two attention mechanisms work together to implement the concept of hierarchical attention.

### 3.2. Text-Enhance Module

After extensive experimentation with remote sensing image caption tasks, this paper discovered that the captions revolve around nouns. If the noun information is incorrect during the caption generation process, the semantics of the entire caption will be changed. In addition, most of the labels for multi-label classification of remote sensing images are the ground object information contained in remote sensing images. Therefore, this paper chooses to enhance the ground object representation of remote sensing images using a multi-label classification task for remote sensing images. This paper uses the word separation tool nltk to separate and lexically annotate reference captions and extract the corresponding nouns in each reference caption. Because each remote sensing image caption corresponds to five reference captions, we use the nouns extracted from the five reference captions as remote sensing image classification labels. In addition, we create a multi-label remote sensing image classification dataset from the entire remote sensing image caption dataset and train a ResNet network-based multi-label image classification model. The entity

features $f_attr$ of image ground objects used in this paper are obtained from a pre-trained multi-label image classification model. The specific extraction process is shown in Figure 3:

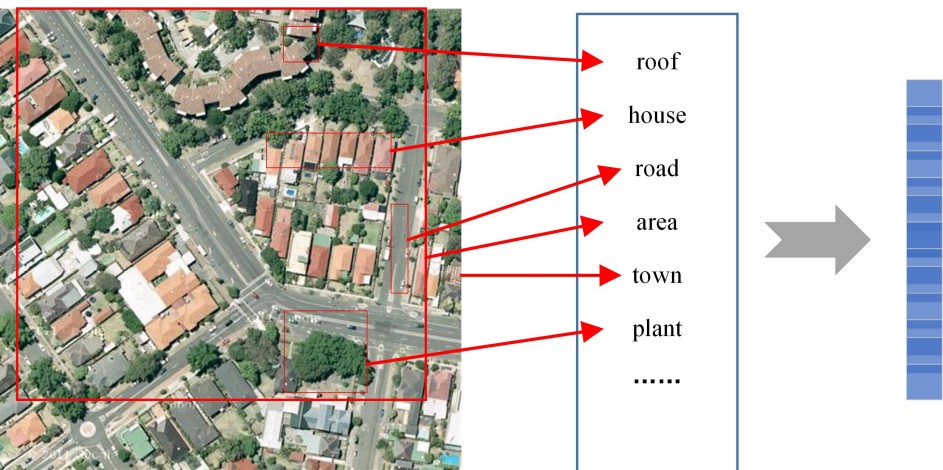

GT：
Lots of houses with different colors of roofs arranged neatly.
A dense residential area with houses arranged neatly and some roads go through this area.
A town with many houses arranged neatly while  the houses are separated by some roads.
A residential area with houses arranged neatly while many plants on the road side.
A dense residential area with houses arranged neatly and some cross roads in this area.

**Figure 3.** Overview of the proposed entity feature acquisition of remote sensing image $f_attr$.

The entity features of the image ground object contain much ground object information, but the text information fed into the model changes constantly. Semantic noise will be generated if we use the entity features of the image ground object. This prevents the ground object from obtaining the correct text information and reduces the model performance. The Text-Enhance module employs a gating mechanism to enrich the acquired entity features of the image ground object $f_attr$ with the ground object information. In this paper, we calculate the weights between entity features of the image ground object $f_attr$ and the current hidden state $h_t^1$ to control the enhancement of ground object information by image ground object features. When $f_t$ is greater than zero, it indicates that the correlation between the entity features of the ground object and the current text information is robust. The model should use more entity features of the ground object to improve the ground object information. Figure 4 shows the Text-Enhance module's structure. The specific process of text guidance features $s_t$ is defined as follows:

$$f_t = (W_f[f_{attr}; h_t^1]) \tag{4}$$

$$st = f_t(W_{fa}f_{tr}) + (1 - f_t)tanh(W_w w_t) \tag{5}$$

$W_f$, $W_f a$, $W_w$ represents the parameters of the model to be learned, and *tanh* represents the activation function. The model can quickly display the corresponding text information of ground objects using st, demonstrating the expression ability of text information of the ground object from the text perspective.

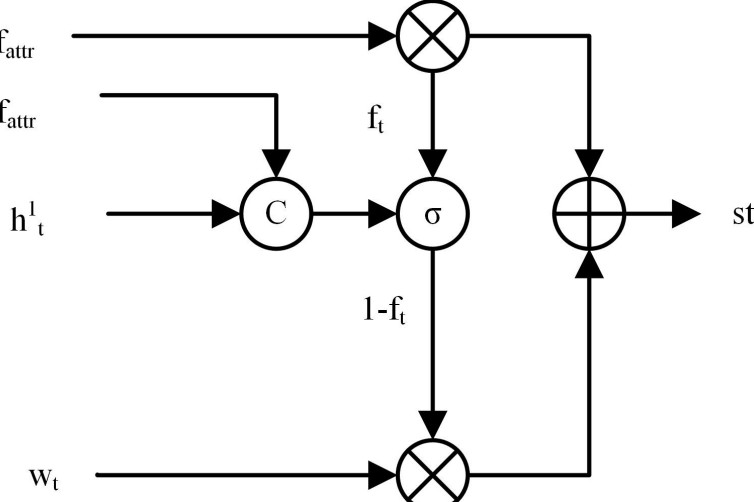

**Figure 4.** Overview of the Text-Enhance module. $f_{attr}$ represents the entity features of the image ground object. $w_t$ represents the word to be input at t times. The Text-Enhance module uses the correlation between $f_{attr}$ and $w_t$ to achieve enhancement of the ground object text information.

## 4. Experiments

### 4.1. Remote Sensing Image Caption Datasets and Evaluation Metrics

#### 4.1.1. Remote Sensing Image Caption Datasets

The UCM-caption dataset [4] contains 2100 images, each with five reference captions. There are 21 scenes in total. This dataset's image size is $256 \times 256$ pixels, and the reference captions are relatively simple, with a slightly monotonous syntax. The Sydney-caption dataset [4] contains 613 images, each with five reference captions. Seven scenes are covered in total. This dataset's image size is $500 \times 500$ pixels, and the reference captions have a longer average length and a more appropriate and richer vocabulary. The RSICD dataset [5] contains 10,921 images, each with up to five reference captions. There are 30 scenes in total. This dataset's image size is $224 \times 224$ pixels. The RSICD dataset captions are relatively complex compared to the UCM-caption and Sydney-caption datasets.

#### 4.1.2. Remote Sensing Image Caption Evaluation Metrics

Five evaluation metrics are commonly used for remote sensing image caption tasks to assess the degree of similarity between model-generated captions and human-annotated captions: BLEU, METEOR, ROUGE, CIDEr, and SPICE.

The BLEU evaluation metric [19] employs $n$-gram matching, which examines how many $n$-gram phrases appear in the reference caption in the generated caption. The general maximum value of $n$-gram is 4. The specific formula is shown below:

$$BP = \begin{cases} 1 & if\ c > r \\ e^{(1-r/c)} & if\ c \leq r \end{cases} \tag{6}$$

$$BLEU = BP \cdot \exp\left(\sum_{n=1}^{N} w_n \log^{p_n}\right) \tag{7}$$

$N$ is the length of the $n$-gram; $w_n$ is its weight, typically $1/N$; $p_n$ is its accuracy rate; $BP$ is the short penalty factor; $c$ is the length of the reference caption; and $r$ is the length of the shortest generated caption. $BLEU$ is implemented based on the accuracy rate, and the larger the value of $BLEU$, the better.

The METEOR evaluation metric [20] employs WordNet to calculate specific sequence matches, matching relationships between synonyms, roots, and affixes, and paraphrases to create a stronger correlation with manual captions, and is based on a harmonic average of

unit group accuracy and recall, with higher METEOR values being better. The particular formula is shown below:

$$F_{mean} = \frac{PR}{\alpha P + (1-\alpha)R}, P = \frac{m}{t}, R = \frac{m}{r} \tag{8}$$

$m$ is the number of $n$-grams that can be matched in the generated caption; $t$ is the length of the generated caption; and $r$ is the number of reference captions. $P$ and $R$ represent accuracy and recall, respectively. A penalty factor $Pen$ is added to penalize cases where the generated caption does not match the reference caption's word order to ensure that more extended utterance matching is performed. The penalty factor is described below:

$$Pen = \gamma \cdot flag^{\beta}, flag = \frac{ch}{m} \tag{9}$$

The word order is based on chunks, and each chunk's matching single words are adjacent and ordered between two strings. The number of chunks is $ch$, and the correlation coefficient is $\gamma, \beta$. The final METEOR score is calculated. The particular formula is shown below:

$$score = (1 - Pen) \cdot F_{mean} \tag{10}$$

The longest common subsequence is denoted by L in the ROUGE-L evaluation metric [21]. The accuracy and recall of the longest common subsequence between the reference caption and the generated caption are calculated to achieve sequential matching of sentence-level word order. The particular formula is shown below:

$$R_{lcs} = \frac{LCS(X,Y)}{m} \tag{11}$$

$$P_{lcs} = \frac{LCS(X,Y)}{n} \tag{12}$$

$$F_{lcs} = \frac{(1+\beta^2)R_{lcs}P_{lcs}}{R_{lcs} + \beta^2 P_{lcs}} \tag{13}$$

$m, n$ are the lengths of the reference caption $X$ and the generated caption $Y$, respectively; $LCS(X,Y)$ is the longest common subsequence of the reference and generated captions; the larger the value of ROUGE-L, the better.

The CIDEr evaluation metric [22] uses TF-IDF (term frequency-inverse document frequency) to assign weights to different $n$-grams. The basic idea is to treat each sentence as a document and then compute the cosine angle of the TF-IDF vector, which yields the similarity between the generated and reference captions. The following is the specific calculation formula:

$$CIDEr_n(c, S) = \frac{1}{M} \sum_{i=1}^{M} \frac{g^n(c) \cdot g^n(S_i)}{\|g^n(c)\| \times \|g^n(S_i)\|} \tag{14}$$

$c$ denotes the generated caption; $S$ denotes the set of reference captions; $n$ denotes the evaluated $n$-gram; and $M$ denotes the number of reference captions, typically five. $g^n(.)$ represents the TF-IDF vector based on the $n$-gram.

The SPICE evaluation metric [23] was chosen to measure the similarity of two captions using the scene graph. It uses the semantic representation of the graph to encode a caption of the objects, attributes, and relationships between them in the utterance and then computes the F-score value of the objects, attributes, and relationships in the generated utterance. The following is the precise formula:

$$SPICE(c,S) = F_1(c,S) = \frac{2 \cdot P(c,S) \cdot R(c,S)}{P(c,S) + R(c,S)} \tag{15}$$

$$P(c,S) = \frac{|T(G(c)) \otimes T(G(S))|}{|T(G(c))|} \tag{16}$$

$$R(c,S) = \frac{|T(G(c)) \otimes T(G(S))|}{T(G(S))} \tag{17}$$

$c$ is the generated caption; $S$ is the set of reference captions; $G(.)$ denotes the conversion of a piece of text into a scene graph using some method; $T(.)$ converts a scene graph into a set of tuples; the $\otimes$ operation is similar to the intersection, differing from the intersection in that it is not strict matching, but similar to matching in METEOR.

### 4.2. Experimental Environment and Parameter Settings

All experiments in this paper were carried out on an NVIDIA GeForce RTX 2080 Ti device. The ratio of data in the training, validation and test sets is 8:1:1 in all three public datasets. In this paper, we acquire ground object information of images using ResNet18, a pre-trained image multi-label classification network. Because the UCM-caption dataset extracts 120 nouns, the Sydney-caption dataset extracts 75 nouns, and the RSICD dataset extracts 1102 nouns, the fully connected layers of the pre-trained image multi-label classification network have dimensions of 120, 75, and 1102. Before being fed into the model, all images are flipped horizontally with a 50% probability. The maximum length of the generated captions is the same as that of the dataset's reference captions. The model employs a pre-trained ResNet18 for the ground object entity of image extraction, with a learning rate of $1 \times 10^{-4}$ and a batch size of 16, and a learning rate decrease of 0.8 every five epochs for a total of 30 epochs. The batch size of the Sydney-caption dataset is set to 8 to avoid overfitting.

### 4.3. Comparison Experiment and Result Analysis

For comparison, we use the following models as comparison models:

- CSMLF [24] model proposes a collective semantic metric learning approach in which five reference captions are transformed into a collective caption.
- Sound-a [16] model uses sound as an additional input to guide the generation of remote sensing image captions. Different sounds of the same image are fed into the trained model, resulting in different captions.
- RTRMN [17] model generates captions with the help of topic word information. During training, the model extracts topic words from captions for remote sensing images.
- SD-RSIC [18] model uses a summary-driven mechanism and an enhanced word vocabulary to provide more detailed captions for semantically complex remote sensing images.
- CapFormer [25] model uses the Swin transformer model as the visual feature processing model and the decoding part of the transformer model as the text generation, avoiding the loss of information from multiple pooling operations.

Tables 1–3 compare the DFEN model to other remote sensing image caption models on various datasets. The best results are highlighted in bold, and the evaluation metrics are taken directly from their papers. The table shows that the current model outperforms the comparison model across all evaluation metrics.

**Table 1.** Results of different models on the UCM-caption dataset.

| METHOD | BLEU-1 | BLEU-2 | BLEU-3 | BLEU-4 | METEOR | ROUGE-L | CIDEr | SPICE |
|---|---|---|---|---|---|---|---|---|
| CSMLF | 0.436 | 0.273 | 0.186 | 0.121 | 0.132 | 0.393 | 0.223 | 0.076 |
| Sound-a | 0.783 | 0.728 | 0.676 | 0.633 | 0.380 | 0.686 | 2.906 | 0.420 |
| RTRMN | 0.803 | 0.732 | 0.682 | 0.639 | 0.426 | 0.773 | 3.127 | 0.453 |
| SD-RSIC | 0.748 | 0.664 | 0.598 | 0.538 | 0.390 | 0.695 | 2.132 | - |
| DFEN (Ours) | **0.851** | **0.784** | **0.728** | **0.677** | **0.459** | **0.805** | **3.177** | **0.501** |

**Table 2.** Results of different models on the Sydney-caption dataset.

| METHOD | BLEU-1 | BLEU-2 | BLEU-3 | BLEU-4 | METEOR | ROUGE-L | CIDEr | SPICE |
|---|---|---|---|---|---|---|---|---|
| CSMLF | 0.600 | 0.458 | 0.387 | 0.343 | 0.248 | 0.502 | 0.756 | 0.262 |
| sound-a | 0.716 | 0.632 | 0.547 | 0.466 | 0.313 | 0.604 | 1.803 | 0.387 |
| SD_RSIC | 0.761 | 0.666 | 0.586 | 0.517 | 0.366 | 0.657 | 1.690 | - |
| DFEN (Ours) | **0.798** | **0.697** | **0.614** | **0.542** | **0.373** | **0.723** | **2.009** | **0.449** |

**Table 3.** Results of different models on the RSICD dataset.

| METHOD | BLEU-1 | BLEU-2 | BLEU-3 | BLEU-4 | METEOR | ROUGE-L | CIDEr | SPICE |
|---|---|---|---|---|---|---|---|---|
| CSMLF | 0.576 | 0.386 | 0.283 | 0.222 | 0.213 | 0.446 | 0.530 | 0.200 |
| sound-a | 0.620 | 0.482 | 0.390 | 0.320 | 0.273 | 0.514 | 1.638 | 0.360 |
| RTRMN | 0.620 | 0.462 | 0.364 | 0.297 | 0.283 | 0.554 | 1.515 | 0.332 |
| SD_RSIC | 0.644 | 0.474 | 0.369 | 0.300 | 0.249 | 0.523 | 0.794 | - |
| CapFormer | 0.661 | 0.499 | 0.400 | 0.326 | - | 0.498 | 0.912 | - |
| DFEN (Ours) | **0.766** | **0.636** | **0.538** | **0.463** | **0.373** | **0.685** | **2.605** | **0.477** |

The CSMLF [19] model has the lowest evaluation metrics of the models discussed above, particularly CIDEr, which are 0.223, 0.756, and 0.530, respectively. The CSML model employs retrieval to obtain captions that correspond to the image's semantic information. While this method reduces syntactic errors in captions, it cannot generate information that does not exist in the retrieval database. The DFEN model fully utilizes deep learning models' inferential learning capability to explore completely new image captions using historical image and text information. The consistency of the captions generated by the DFEN model is improved when compared to the CSMLF model.

The Sound-a [16], RTRMN [17], and SD-RSIC [18] models use sound, topic words, and summary information to enhance ground object information. External information is sound and slightly less relevant to visual information than topic words and summaries. Although the summary information contains more textual content than the topic word information, it is only textually enhanced and contains less useful information. Therefore, the sound-a [16] and the SD-RSIC [18] models have the lowest evaluation metrics among the three models mentioned above that use additional information. Remote sensing images contain much information about ground objects. The semantic information expressed by a single topic word is slightly weaker than multiscale image context features and image multi-labeling information. The DFEN model uses multiscale image information to enhance the visual semantic information of remote sensing image ground objects and enriches ground object text information using an image multi-label classification task and a gated fusion mechanism. This reinforces the need for additional information to be added to ground object information. The DFEN model improves ground object information from image and text perspectives, resulting in better performance than the previous models, which only consider image or text information. Furthermore, the CapFormer [25] model's evaluation metrics in the RSICD dataset are significantly lower than those of the DFEN model. The characteristics of the transformer model and CNN models' characteristics are primarily responsible for this. The Transformer model does not consider the images' inductive bias information but compensates for it by training a large amount of data. The remote sensing image caption dataset, on the other hand, contains fewer data. As a result, the CapFormer model's evaluation metrics are significantly lower than those of the DFEN model for the

same dataset. As a result, we can conclude that DFEN improves the images and text information of ground objects. Furthermore, the image context vector proposed by the DFEN model contains information about the image's multiscale ground object features, which gives the model an excellent inductive bias.

### 4.4. Ablation Experiment and Result Analysis

We created the ablation model to demonstrate the superiority of the model's key components. The Baseline model represents the baseline model, the Baseline-v model represents the Baseline model with the Image-Enhance module added, and the Baseline-st model represents the Baseline model with the Text-Enhance module added. Table 4 displays the ablation model's performance on the three datasets. The visual characteristics obtained by the two ablation models are too abstract. As a result, we conducted experimental analysis by evaluation metrics for the ablation model's caption. According to the results, both the Image-Enhance and the Text-Enhance modules proposed in this paper improve model performance. First, $v$ is image context features produced by Image-Enhance that incorporates multiscale image features. Each evaluation metric is slightly improved after using $v$ compared to the Baseline. However, the improvement of BLEU1-4 is less than 1%, indicating that the model is ineffective in generating the same $n$-grams as the reference captions. Furthermore, the Baseline-v model obtains more ground object information from remote sensing images than the Baseline model, which only uses single-layer image semantic features by using multiscale image features generated by different convolutional layers. This results from the image context feature enhancing ground objects' visual features in remote sensing images and making it easier for the model to generate diverse captions. Second, $st$ is Text-Enhance generated text guidance features that incorporate additional visual information from remote sensing images. The evaluation metrics on each dataset are improved after using $st$. This implies that the additional entity features of the image ground object supplement the visual information and broaden the textual information space. This is primarily because the additional entity features of the image ground object supplement the visual features of the ground object information in the remote sensing image. They also filter out irrelevant visual features by fusing them with textual information. Table 4 and Figure 5 show that the DFEN model's dual features can achieve a more accurate remote sensing image caption.

**Table 4.** Results of ablation experiment based on different remote sensing image caption datasets.

| DataSet | METHOD | BLEU-1 | BLEU-2 | BLEU-3 | BLEU-4 | METEOR | ROUGE-L | CIDEr | SPICE |
|---------|--------|--------|--------|--------|--------|--------|---------|-------|-------|
| UCM | Baseline | 0.765 | 0.678 | 0.619 | 0.573 | 0.394 | 0.724 | 2.678 | 0.444 |
| | Baseline-v | 0.770 | 0.685 | 0.627 | 0.579 | 0.406 | 0.735 | 2.736 | 0.472 |
| | Baseline-st | 0.833 | 0.759 | 0.701 | 0.651 | 0.456 | 0.800 | 3.168 | 0.494 |
| | DFEN | 0.851 | 0.784 | 0.728 | 0.677 | 0.459 | 0.805 | 3.177 | 0.501 |
| Sydney | Baseline | 0.775 | 0.663 | 0.578 | 0.506 | 0.358 | 0.700 | 1.862 | 0.415 |
| | Baseline-v | 0.784 | 0.673 | 0.583 | 0.508 | 0.356 | 0.713 | 1.939 | 0.438 |
| | Baseline-st | 0.780 | 0.683 | 0.610 | 0.553 | 0.368 | 0.715 | 1.979 | 0.450 |
| | DFEN | 0.798 | 0.697 | 0.614 | 0.542 | 0.373 | 0.723 | 2.009 | 0.449 |
| RSICD | Baseline | 0.715 | 0.576 | 0.477 | 0.402 | 0.343 | 0.646 | 2.133 | 0.427 |
| | Baseline-v | 0.727 | 0.584 | 0.484 | 0.410 | 0.334 | 0.643 | 2.184 | 0.422 |
| | Baseline-st | 0.758 | 0.626 | 0.526 | 0.45 | 0.359 | 0.679 | 2.566 | 0.478 |
| | DFEN | 0.766 | 0.636 | 0.538 | 0.463 | 0.373 | 0.685 | 2.605 | 0.477 |

In order to demonstrate the effectiveness of the method proposed in this paper more visually, we show the generated results of some of the test data. To emphasize the experiments' effectiveness, we highlighted the words in the Baseline model captions that do not match the semantic information of the remote sensing images in bold blue font and the differences between the DFEN model and the Baseline model in bold red font.

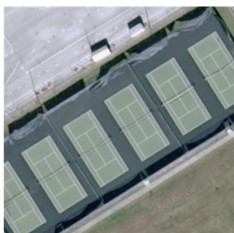

**GT:**
Many tennis courts arranged in line with some plants beside .
Many tennis courts are arranged in line and surrounded by lawn .
There are many tennis courts arranged in line with lawn beside .
There are many tennis courts arranged in line and surrounded by lawn .
There are many tennis courts arranged in line with lawn beside .
**Baseilne:**
Two tennis two are surrounded by some plants .
**DFEN:**
Some tennis **courts** arranged neatly and surrounded a **road** .

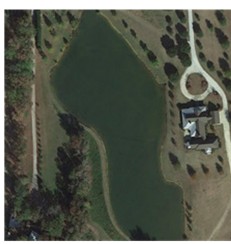

**GT**:
a villa is located in the wilderness .
green vegetation .
some green trees are near an irregular pond .
some green trees are near an irregular pond .
a villa is located in the wilderness .
**Baseline:**
a building is surrounded meadows .
**DFEN:**
several **green trees** and meadows are in two sides of a **green river** .

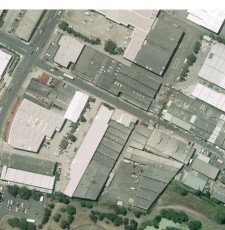

**GT:**
An industrial area with many white and grey buildings densely arranged while a lawn beside .
There is a lawn with a industrial area beside .
Many white and grey buildings arranged densely in the industrial area while a lawn beside .
**Baseline:**
A residential area goes through houses industrial area with some white buildings and **a parking lot** beside .
**DFEN:**
Some **roads** go through some **industrial area** with many **buildings** and **warehouses** .

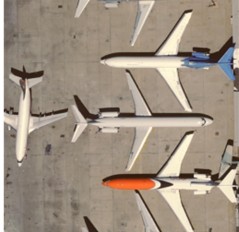

**GT:**
Four different airplanes are stopped dispersedly at the airport .
Four airplanes scattered at the airport .
Four airplanes are stopped dispersedly at the airport .
Many different kinds of airplanes are stopped at the airport .
There are many airplanes at the airport .
**Baseilne:**
An airplane with **purple** fuselage is stopped at the airport .
**DFEN:**
An airplane with **blue** fuselage is stopped at the airport .

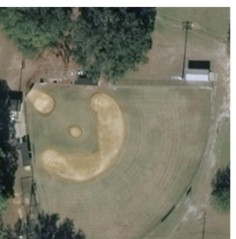

**GT:**
A regular baseball diamond compose of manicured lawns and sand .
A regular baseball diamond compose of manicured lawns and sand .
It is a regular baseball diamond .
It is a small baseball diamond with sands and grass .
This is a baseball diamond .
**Baseilne:**
A regular baseball diamond compose of **houses** and plants .
**DFEN:**
A regular baseball diamond compose of **manicured lawns and sand** .

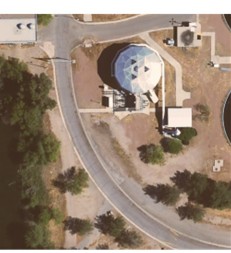

**GT:**
A storage tank is on the ground with some houses and a road beside .
There is a white storage tank on the ground with some houses and a road beside .
A storage tank is on the ground with some houses and a road beside .
There is one white storage tank on the ground with some houses and a road beside .
A storage tank is on the ground with some houses and a road beside .
**Baseilne:**
A storage tank is on the lawn and some **cars** parked beside .
**DFEN:**
A storage tank is on the lawn and some **houses** beside .

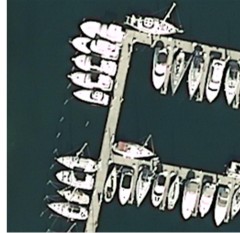

**GT:**
Lots of boats docked neatly at the harbor .
Many boats docked neatly at the harbor and the water is deep blue .
Many boats docked neatly at the harbor and only a few positions are free .
Lots of boats docked neatly at the harbor and only a few positions are free .
Lots of boats docked at the harbor and the water is deep blue .
**Baseilne:**
A boats docked neatly at the harbor and some positions are free .
**DFEN:**
**Lots of** boats docked neatly at the harbor and some positions are free .

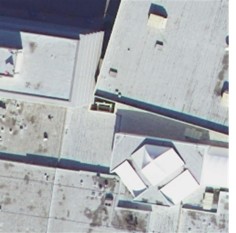

**GT:**
Some buildings are pressed together .
Some buildings with white and grey roofs are pressed together .
There are some buildings pressed together .
Some buildings with white and grey roofs .
There is are some buildings with white and grey roofs .
**Baseilne:**
Buildings and **plants** .
**DFEN:**
Some buildings **with white roofs** .

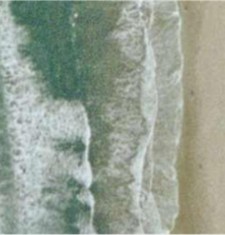

**GT:**
Waves are slapping the white sand beach and throwing up white foams .
Waves beat on the beach over and over again .
Waves are slapping a white sand beach over and over again and throw up white foam .
Waves are slapping a white sand beach while some birds flying .
Waves are slapping the white sand beach and throwing up white foams .
**Baseilne:**
This is a beach with violent waves .
**DFEN:**
Waves come to the **white sand** beach **over and over again with white foams** .

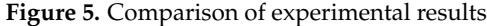

**Figure 5.** Comparison of experimental results.

Figure 5 shows the test results of the remote sensing images under the DFEN model. From the third, fourth, fifth, sixth, and eighth subfigures, we can see that the captions

generated by the Baseline model all show words that do not match the semantic information of the remote sensing images. For example, in the airplane scene, we can see that the plane is blue or yellow rather than purple; in the storage tank scene, the Baseline model misidentifies the white house as a car due to the small scale of the feature; and in the building scene, the Baseline model misidentifies the grey object as a plant because the grey feature is closer to the plant in the remote sensing image. This is because the Baseline model only considers the single-scale features of the image. In contrast, the Image-Enhance module of the DFEN model employs a hierarchical attention mechanism to obtain richer multiscale features, which not only supplement the neglected small-scale ground object information but also alleviate the problem of inaccurate ground object identification due to variable ground object scales. As a result, any incorrect semantic information in the Baseline model's captions is corrected in the DFEN model.

Furthermore, in the first, second, seventh, and ninth subfigures of Figure 5, we can see that the captions generated by the DFEN model have added some new content in line with the semantic information of the remote sensing images when compared to the captions generated by the Baseline model. For example, in the tennis scene, the DFEN model generates captions with more detail for "tennis", "plants" and "arranged nearly"; in the boat scene, the DFEN model generates more detailed "Lots of"; in the beach scene, the DFEN model generates "white sand" and "Over and over again with white foams'. These are the results of ground object information enhancement by text-guided vectors in the Text-Enhance module of the DFEN model. The entity features of the image ground object $f_{attr}$ enrich the relevant ground object information in the remote sensing image by fusing it with textual information. As a result, the DFEN model generates captions that contain more information about the ground objects of remote sensing images. The Image-Enhance module generates image context features $v$ sing multiscale image information to supplement neglected ground object information. The Text-Enhance module generates text-guided features classification labels of remote sensing images to direct attention to more text-related ground object information. This allows the improved ground object information to be more closely related to semantic information, resulting in more accurate remote sensing image captions. This results from the DFEN model's Image-Enhance and Text-Enhance modules working together.

### 4.5. Setting and Analysis of Experimental Parameters

The batch size determines the model training's computational efficiency. Due to this paper's complexity of the model structure, we chose to train the model with batch sizes of 8, 16, and 32 to achieve optimal model performance. Table 5 displays the training time of the model with various batch sizes. Figure 6 depicts the model's training results with various batch size settings. When batch size is increased from 8 to 32, the DFEN model gradient decreases faster and converges better. When the batch size was 8, the value of bleu1 was higher in the first 15 epochs. When the batch size was 16, the value of bleu1 was higher in the previous 15 epochs. Furthermore, at batch size 16, the DFEN model has the shortest training time. As the batch size increases, the time required to train each epoch is minimized, and the loss value decreases more slowly. The optimal time is reached when the batch size is increased to a specific value. In order to balance the different evaluation metric values with the training time, we finally set the batch size to 16.

**Table 5.** Training time of DFEN with different batche_size in UCM-caption dataset.

| DataSet | Batch_Size | Training Time |
|---------|-----------|---------------|
| UCM | 8 | 15.37 |
| | 16 | 13.87 |
| | 32 | 15.93 |

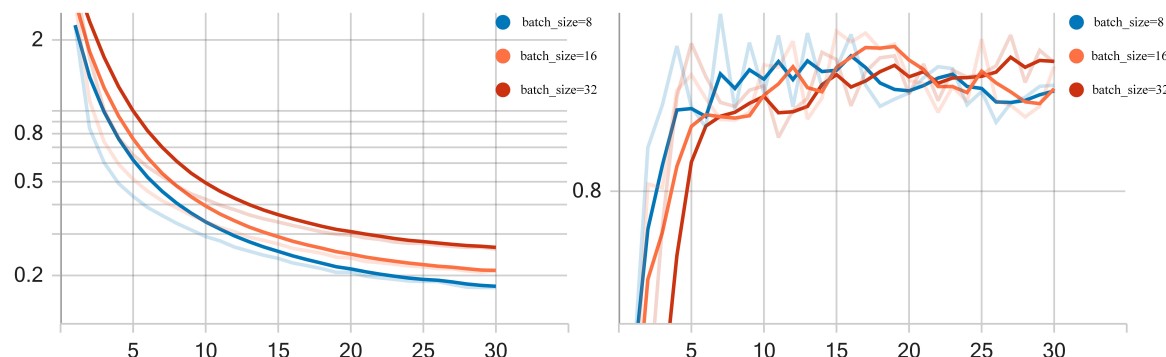

**Figure 6.** Convergence and Bleu-1 values of DFEN Model with different batch_size in UCM-caption dataset.

The DFEN model was chosen to use the ResNet model as the backbone model for image visual feature extraction and multi-label classification of remote sensing images. Therefore, the network structure and the number of parameters of the ResNet model also affect the experimental results of the DFEN model. In this paper, we choose to use ResNet18, ResNet34, ResNet50 and ResNet101 models, which are commonly used in image classification tasks, for training. Table 6 shows the comparison of the evaluation metrics of different ResNet models on the UCM dataset. Through the experiments we can find that the evaluation metrics of the DFEN model gradually improve with the increase of the Resnet network model parameters as well as the network depth in the models of ResNet18-50, especially the evaluation metric SPICE. By calculating the similarity between the reference and the generated caption, the SPICE evaluation metric compares the differences between the reference and the generated caption. This implies that the ResNet model's unique residual structure not only mitigates the problem of model degradation, but also generates more detailed local features that enrich the ground object information in the captions. However, when compared to the ResNet50 model, the evaluation metrics produced by the ResNet101 model do not improve. The images in the UCM dataset are mostly airport scenes. In the airport scenes, the image features have less information. This results in the ResNet101 model, which has more network depth and a higher number of parameters, being less effective than ResNet50. In addition, the increase in the number of network layers and network parameters also generates a large amount of computation, resulting in slow training and loading of the model. Therefore, in order to balance the network parameters and computational cost, the Resnet18 model is chosen as the visual feature extractor and label acquisition model for the DFEN model.

**Table 6.** Evaluation metrics results of DFEN with different ResNet in UCM-caption dataset.

| Model | BLEU-1 | BLEU-2 | BLEU-3 | BLEU-4 | METEOR | ROUGE-L | CIDEr | SPICE |
|---|---|---|---|---|---|---|---|---|
| ResNet18 | 0.851 | 0.784 | 0.728 | 0.677 | 0.459 | 0.805 | 3.177 | 0.501 |
| ResNet34 | 0.856 | 0.786 | 0.727 | 0.671 | 0.458 | 0.813 | 3.176 | 0.510 |
| ResNet50 | 0.865 | 0.803 | 0.746 | 0.700 | 0.462 | 0.807 | 3.224 | 0.524 |
| ResNet101 | 0.862 | 0.797 | 0.742 | 0.694 | 0.453 | 0.805 | 3.194 | 0.498 |

Image understanding and the interaction between images and text are critical to the current remote sensing image caption. Image understanding involves not only using image representation features but also using underlying information to enrich the visual representation of the image. Image and text are two different vector spaces. The interaction between image and text determines the expression ability of text. To fuse features of different scales of ground objects, Image-Enhance employs a hierarchical attention mechanism. The image multi-label classification task extracts the potential information of the image. These make the visual representation of ground object information adequate. The Text-Enhance module's gating mechanism uses image interaction to enrich ground object information.

The DFEN model improves the performance of the DFEN by improving remote sensing image information in terms of image understanding and graphical interaction.

## 5. Conclusions

To address the problem of inadequate ground object information acquisition, this paper proposes a dual feature enhancement network model DFEN that enhances ground object information from both image and text perspectives. To supplement the neglected ground object information, the DFEN model employs a hierarchical attention mechanism to obtain multi-scale semantic information from remote sensing images. To obtain additional ground object entity features, the DFEN model employs a pre-training model for multi-label classification of remote sensing images, which is combined with a gating mechanism to obtain correct ground object information. The DFEN model improved the BLEU by at least 10.5%, the ROUGE-L by 18.7%, and the CIDEr by 1.09 when compared to the CapFormer and RTRMN models. This indicates that the DFEN model's multi-scale semantic information and the additional visual information from remote sensing images improved the accuracy of remote sensing image caption generation. In future work, we will explore the impact of more additional visual information and multi-scale visual backbone on remote sensing image caption tasks in order to obtain more semantic information of remote sensing images and generate more accurate remote sensing image captions. Furthermore, the remote sensing image caption dataset is small and biased. In future work, we may choose to use simulated data [26,27] to create remote sensing image caption datasets for different regions and scenarios, thereby improving the model's generalisation capability. Furthermore, manual confirmation of the condition of high-resolution remote sensing images captured by UAVs is still required in the field of UAV inspection. In the future, we will be able to automatically generate inspection reports for high-resolution remote sensing images captured by UAV inspections, freeing up manual labor and improving UAV inspection efficiency.

**Author Contributions:** Conceptualization, W.Z.; methodology, W.Z.; validation, W.Z.; writing—original draft preparation, W.Z.; writing—review and editing, W.Z., W.Y. and D.C.; supervision, W.Y. and F.W.; funding acquisition, W.Y. All authors have read and agreed to the published version of the manuscript.

**Funding:** This research was funded by the Natural Science Foundation of China grant number 202204120017, the Autonomous Region Science and Technology Program grant number 2022B01008-2, and Autonomous Region Science and Technology Program grant number 2020A02001-1.

**Data Availability Statement:** The data used in this study are public datasets, which can be obtained from the following link: https://github.com/201528014227051/RSICD (accessed on 24 March 2023).

**Conflicts of Interest:** The authors declare no conflict of interest.

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
