# Peer review of "DFEN: Dual Feature Enhancement Network for Remote Sensing Image Caption"

_electronics, doi:10.3390/electronics12071547_

Round 1

Reviewer 1 Report

In this paper, the authors proposed a dual feature enhancement network for Remote Sensing Image Caption. The proposed model contains sufficient novelty and results are efficient in comparison of existing techniques. There are some suggestions:

1. Caption of the figure 1 is mixed with the text. 

2. Replace * by x in equation(2)

3. In equation (6) and (7), pass parameters to BP.

4. Rewrite the conclusion section by including improvement data in percentage in comparison of recent existing techniques.

Reviewer 2 Report

The presented paper is well organized and presents very good results in terms of image caption generation. In order to improve quality of results please refer following remarks:

- What is the relation in the number of samples used for training, validating and testing?

- Please try to improve the quality of figure 6.

- What are the potential practical applications and your further plans in that area - improve final sentence in conclusions part.

- The application of simulated data (from computer games environment) is getting much more popular. Please comment in your work if such an application could be fitted similarly in evaluation and validation process (example https://doi.org/10.3390/s20174960) and training mode (https://doi.org/10.3390/s20185250) for improvement of final results. In such an approach one could simulate different cities or areas with different weather/light conditions and try to improve image captioning.
